# Influence of the Menstrual Cycle on Performance in CrossFit®

**Jeimy S. Mora-Serrano [1], Iván Chulvi-Medrano [2], Ismael Martínez-Guardado [3] and Diego A. Alonso-Aubin [3,\*]**

[1] Faculty of Health Sciences—HM Hospitals, University Camilo José Cela, C/Castillo de Alarcón, 49, 28692 Madrid, Spain; jeimystephani.mora@alumni.ucjc.edu

[2] Research Group in Prevention and Health in Exercise and Sport (PHES), Department of Physical Education and Sports, Facultad de Ciencias de la Actividad Física y el Deporte, Universidad de Valencia, 46010 Valencia, Spain; ivan.chulvi@uv.es

[3] Strength Training and Neuromuscular Performance Research Group (STreNgthP), Faculty of Health Sciences—HM Hospitals, University Camilo José Cela, C/Castillo de Alarcón, 49, Villanueva de la Cañada, 28692 Madrid, Spain; imartinez@ucjc.edu

\* Correspondence: diegoalexandre.alonso@ucjc.edu; Tel.: +34-690-093-962

**Abstract:** The purpose of this study was to compare the fast maximal dynamic strength, maximal dynamic strength, and muscular endurance changes between the follicular phase (FP) and luteal phase (LP) in the menstrual cycle (MC). Fourteen healthy resistance-trained CrossFit® athletes and eumenorrheic women without oral contraception participated in this study (age: 29.64 ± 3.95 years; height: 1.62 ± 0.04 m and mass: 60.43 ± 6.56 kg). A double cross-sectional and descriptive study was developed to evaluate strength, power, and muscular endurance performance in the FP (days 5 to 7) and LP (days 20 to 23). In each assessment, the subjects performed three countermovement jumps (CMJs) for assessing fast maximal dynamic strength using My Jump 2, one-repetition maximum (1-RM) on deep squat for maximal dynamic strength, and the Karen® protocol for muscular endurance. The data were analyzed using a paired sample t-test to determine whether there were any significant differences between FP and LP for all the assessments and the significance was set at $p \le 0.05$. Results revealed no significant differences between the FP and LP in performance: (a) fast maximal dynamic strength (FP: 0.27 ± 0.04 m; LP: 0.29 ± 0.04 m; t = 0.98; $p = 0.34$; ES = 0.26); (b) maximal dynamic strength (FP: 88.14 ± 12.60 kg; LP: 87.07 ± 12.51 kg; t = 0.90; $p = 0.38$; ES = 0.24); and (c) muscular endurance (FP: 459.43 ± 92.71 s; LP: 456.93 ± 110.68 s; t = 0.27; $p = 0.78$; ES = 0.07). CrossFit® performance is not altered during the MC in CrossFit® trained athletes. However, monitoring the MC and its symptoms could be used to adapt the training prescription to the needs and preferences of the athletes.

**Keywords:** follicular phase; luteal phase; training; strength; female; response

## 1. Introduction

Research studies on sports performance have increased in recent years, but there is a gap as females represent only 34% of the athletes studied, with only 6% of total publications being conducted exclusively on females. Thus, they remain underrepresented within sport and exercise science research [1]. Furthermore, it has traditionally been assumed that physiological responses to training and exercise are similar in men and women, so the results obtained in men have been used to be extrapolated to women [2]. The non-inclusion of female athletes has been justified by the methodological challenges with studying their hormonal status, including the menstrual cycle (MC) [3,4].

However, there is an increasing interest in female training and the MC because of the potential for optimizing athletic performance through understanding hormonal fluctuations, managing menstrual symptoms, and strategically timing training to phases of the MC for better adaptation and recovery. The MC's average length is 28 ± 2.4 days and may be divided into four phases, which are regulated by hormonal changes in the following

order: menstruation, follicular phase (FP), ovulation, and luteal phase (LP). The FP lasts 12–14 days and shows low levels of estrogen and progesterone, ovulation takes 1 day and is preceded by an estrogen surge, and the LP lasts 12–14 days and is characterized as displaying high levels of estrogen and progesterone [5]. In fact, hormones show changes in concentrations over the MC and can have different target organs for promoting different physiological end points [6].

During the MC, the fluctuations in hormone concentrations through the different phases could lead to differences in neuromuscular performance and strength or aerobic training adaptations [7,8]. For example, estrogen augments our glycogen storage capacity and free fatty acid availability as a fuel source, and this could lead to decreased carbohydrate use or glycogen sparing [9]. In addition, high estrogen levels can be associated with enhanced gluconeogenesis and a greater uptake of glucose into type I fibers [10]. Consequently, a greater oxidative capacity is associated with lower blood lactates levels and time to exhaustion [11]. On the other hand, progesterone increases our heart rate, ventilation, and body temperature, which could lead to thermoregulatory alterations, increasing the subjective perception of greater exertion and causing decreased athletic performance [12,13].

A recent review has reported that responses to physical training can be different throughout the MC as exercise performance might be reduced during the early FP compared to the LP, and this could lead to different adaptations [14].

Research suggests that for resistance training adaptations in muscle mass and strength, FP-based training could lead to greater outcomes compared to LP-based training. However, there is no strong evidence supporting this information, and more studies are needed [15]. In this sense, a review of this information indicates that the early FP is unfavorable for all strength classes and peak performance in isometric strength is seen in the late FP, whereas isokinetic strength peaks during ovulation. In addition, the same study explains that dynamic strength is optimal in the late FP [1,16].

One possible explanation for this is that the results reported by other studies where trained and untrained females showed different strength and power performance metrics over the course of the MC [17]. In addition, there is interindividual variability in the MC, with disturbances (e.g., amenorrhea, irregular menstruation, pain, etc.) that could explain the differences in physical performance and perceptions [18,19]. While some female athletes feel a decrease in their physical capacity, performance, and rate of perceived exertion over the different phases of the MC, others do not have the same perception [20].

There are contradictions in the scientific literature since increases in muscle strength in the FP and a decrease in fatigue have been reported [21], while other studies have not reported significant differences in muscle strength values between the different phases of the MC [22,23]. In the case of aerobic or anaerobic training, there is no evidence that performance fluctuates over the duration of the MC [24].

Some studies suggest that hormonal fluctuations during the MC can have an impact on neuromuscular performance and muscle strength [25], while other studies claim that these fluctuations do not affect muscle contractile characteristics [26]. However, most studies do not show clear evidence to support these statements [26–28].

Despite all the implications that the MC can have in the prescription of training and its adaptations, there is no clear evidence on the parameters that should be taken as most relevant. Consequently, it is important to expand the research on the effects of the different phases of the MC regarding training adaptations and neuromuscular performance fluctuations in female athletes [29].

In this sense, we believe that the use of a trained sample can enhance our knowledge about the influence of the menstrual cycle on female performance as this population group is more consistent in its performances, regardless of internal or external influences. In addition, to the best of our knowledge, there are no studies on the sports performance and the MC's effects in female CrossFit® athletes.

CrossFit® athletes need good levels of fast maximal dynamic strength (force produced maximally against no or little additional load over quick movement times, e.g., CMJ),

maximal dynamic strength (maximal external resistance an athlete can lift during a full voluntary contraction, e.g., 1 RM), and muscular endurance (the ability to repeatedly exert force against a form of resistance, e.g., Karen® protocol) to achieve good performance.

Consequently, to address gaps in the literature as well as the performance in strength, power, and muscular endurance differences between the FP and LP during the MC, the main objective of this study is to analyze the fast maximal dynamic strength, maximal dynamic strength, and muscular endurance in trained Crossfit® athletes in different MC phases. It is hypothesized that there are performance differences between the FP and LP.

## 2. Materials and Methods

### 2.1. Participants

Fourteen healthy resistance-trained CrossFit® athletes and eumenorrheic women without oral contraception (age: 29.64 ± 3.95 years; height: 1.62 ± 0.04 m and mass: 60.43 ± 6.56 kg) volunteered to participate in this study. To be able to participate in this study, they also had to meet the following inclusion criteria: (1) have more than 1 year of experience in CrossFit®, (2) performed strength training and CrossFit® at least three times a week in the last three months, and (3) can properly perform a squat with good technique, following NSCA guidelines. The exclusion criteria were (1) not having a regular menstrual cycle, (2) have some type of muscular or joint discomfort, (3) absence at the test on the designated day, (4) experienced any medical condition, illness, or injury during the 28 days of this study, and (5) use oral contraceptives.

Participants were informed about the purpose, procedures, and risks of this study, and an informed consent was obtained from each participant. All procedures of this study were performed in accordance with the Declaration of Helsinki and approved by the Ethics Committee of the Camilo José Cela University, Spain.

### 2.2. Study Design and Procedures

A double cross-sectional and descriptive study was developed to evaluate the fast maximal dynamic strength, the maximal dynamic strength, and muscular endurance in the FP and LP. Participants were tested in a well-prepared and conditioned Crossfit® box and by an experienced coach with ten years of experience in strength and conditioning assessments. The entire study for each participant took 28 days, and tests were conducted between days 5 and 7 for the FP and between days 20 and 23 for the LP of the MC. We ensure that participants did not perform any high-intensity physical activity or training 48 h before the assessments. In addition, all the participants performed the same training sessions and lifted the same loads during this study. We did not control their diets during this study, but participants were instructed to not change their normal diet pattern. In addition, during the whole length of this study, ergogenic (e.g., creatine) or stimulant aids (e.g., caffeine) were not allowed.

Firstly, all participants completed a familiarization session with the protocol to ensure appropriate technique within the tests was used. The following protocol was used for the FP and LP (Figure 1): standardized warm-up preceding the assessments comprising (1) 5 min of mobilization and dynamic exercises of the ankles, knees, hips, and shoulder girdle joints; (2) two rounds of a dynamic warm-up consisting of 10 v-ups, 10 supermans, and 10 tempo squats with 1 min of recovery between rounds.

Performance assessments were completed in the following order and included (1) fast maximal dynamic strength exercises that consisted of executing 3 rehearsal countermovement jumps (CMJs) with technical instructions and performing 3 CMJ attempts that were recorded with the app My Jump 2; (2) maximal dynamic strength assessment that consisted of performing a one-repetition maximum (1-RM) on a deep squat, and (3) a muscular endurance assessment with the CrossFit® benchmark Karen®. The three tests were carried out at the same time of the day (18:00–20:00), and there was 5 min of recovery between them.

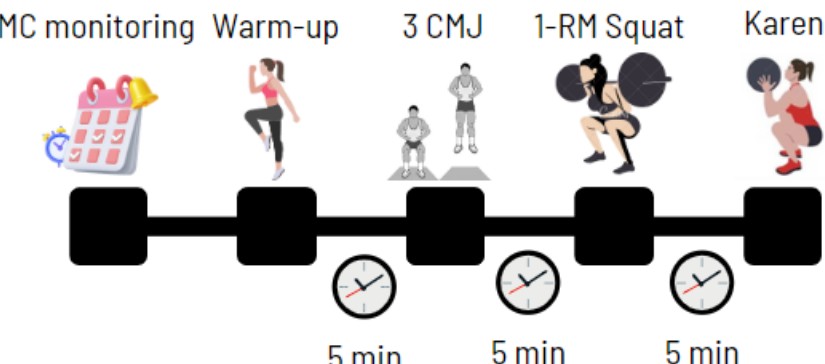

**Figure 1.** Schematic description of the protocol.

- Monitoring of menstrual cycle

The MyCalendar® application (v1.746.280, Period-tracker, Edgewater, MD, USA) was used to monitor participants' MCs. MyCalendar® is one of the most downloaded applications and used by women to monitor their MCs [30]. The application allows users to observe the relationship between their daily calendar and their MCs, with the aim of being able to determine the days corresponding to the FP and LP. During the previous month and the month of this study, participants monitored their MCs using the application to determine the specific days of the assessments and data collection. An experienced researcher instructed the participants about the use of the application and answered all their questions.

- Fast maximal dynamic strength

Fast maximal dynamic strength was determined with 3 CMJs recorded with the app My Jump 2 (v1.0.8). My Jump 2 is a reliable and valid tool for measuring different jump actions in trained female athletes [31]. To record each jump's performance, an iPhone 13 (v15.0.1, Apple, Inc., Cupertino, CA, USA) was used, which includes a 240 Hz high-speed camera at a quality of 1080 p. The researcher laid prone on the ground with the iPhone 13, facing the participant at 1.5 m and zooming in on their feet [32]. With hands on their hips (akimbo position) and starting from a static position, participants were instructed to jump as fast and as high as possible, maintaining their legs straight during the flight phase. For data collection, the first frame in which both feet were off the ground (take-off phase) and the first frame in which at least one foot was touching the ground (start of landing phase) were recorded. Participants performed 3 practice CMJs and then 3 recorded CMJs to ensure proper technique (Figure 2). Jump height (m) and mean values of velocity (m/s), force (N) and power (W) of each CMJ were recorded. The average value of the 3 CMJs for each variable was used for statistical analysis (Figure 2).

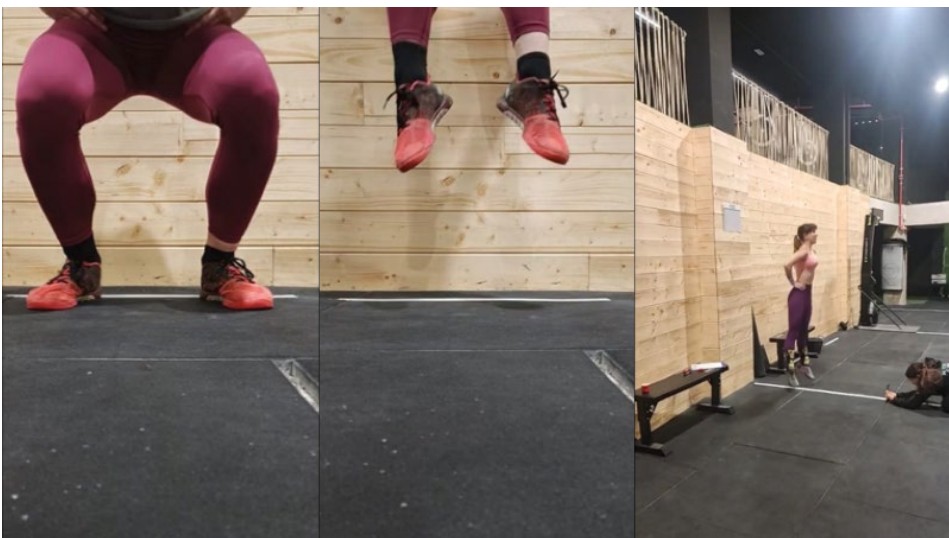

**Figure 2.** Fast maximal dynamic strength (CMJ).

First frame: eccentric phase. Second frame: concentric phase. Third frame: researcher's position during the execution of the CMJ.

- Maximal dynamic strength

Maximal dynamic strength was determined using a one-repetition maximum deep back squat (1-RM) in kilograms (kg), which is highly reliable and can be used effectively [33]. The standardized protocol of the National Strength and Conditioning Association (NSCA) was used [34]. Participants were permitted to a maximum of 6 progressively increasing loads with 3 min rest period between attempts, and all participants completed the test within 4–5 attempts. During the 1-RM attempt, only if the participant achieved to lower the thighs to a position at least parallel to the floor (90° knee angle), their attempt was classed as valid. If not, the lift was performed again after a 3 min rest period. During the 1-RM lifts, all participants were encouraged equally, and technical instructions were provided to ensure correct knee flexion of at least 90°. In addition, a second researcher was filming the lifts to ensure proper knee flexion for a deep squat (Figure 3).

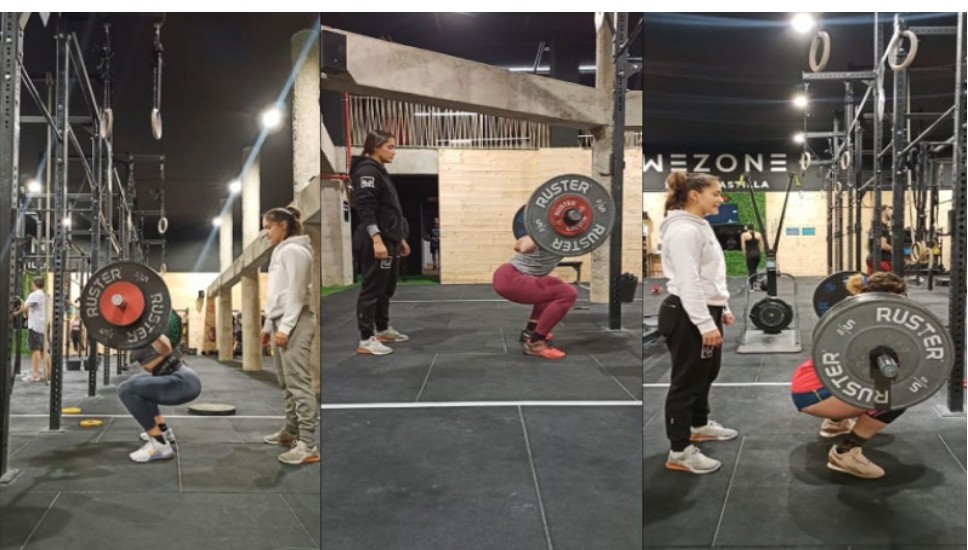

**Figure 3.** Maximal dynamic strength (1-RM deep back squat).

- Muscular endurance

Muscular endurance was tested individually using a CrossFit® benchmark named Karen®. The Karen® workout consists of throwing a 6 kg medicine ball 150 times against the wall above a standard height of 2.74 m (Figure 4). All the participants were familiar with the test and knew the task. However, they were reminded of its key aspects so that each repetition was counted: (1) bend the knee to lower the hips below parallel to the floor and (2) touch the wall with the ball above a minimum height of 2.74 m. The time spent in seconds (s) by each participant for completing the Karen® protocol was recorded with the iPhone 13 chronometer (Figure 4).

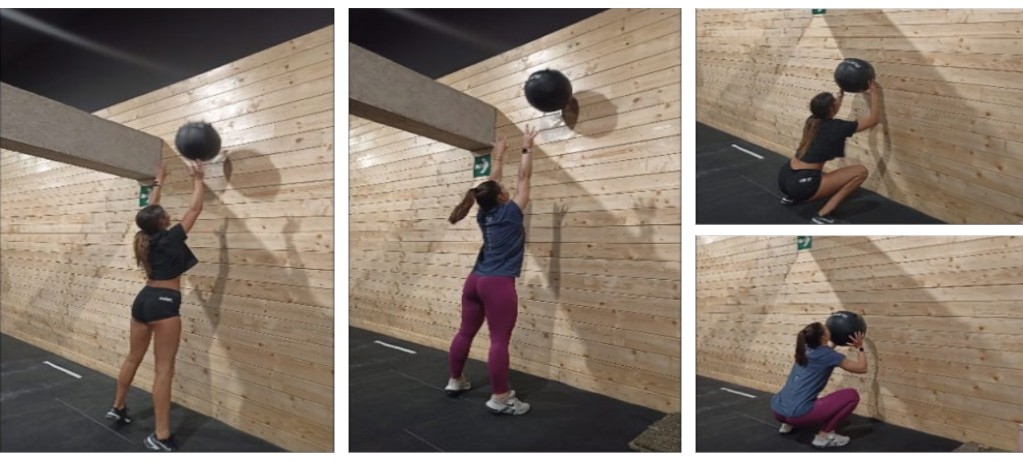

**Figure 4.** Muscular endurance (Karen®). Karen protocol where the athlete performed a throw and catch with one deep squat with a 6 kg medicine ball.

*2.3. Statistical Analyses*

Data are presented as mean ± standard deviation (SD). For testing the normality of the data, a Shapiro–Wilk test was conducted. A paired sample t-test was conducted to determine whether there were any significant differences between the FP and LP for all the performance assessments, and the significance was set at $p \leq 0.05$. Effect sizes (ESs) were determined using Cohen´s d method [35] and interpreted based on the recommendations of Rhea, which defines <0.35, 0.35–0.80, 0.8–1.5, and >1.5 as trivial, small, moderate, and large, respectively [36]. Power analysis revealed that a sample size of 14 can reliably (with probability greater than 0.32) detect effect sizes d ≥ 0.32, assuming a one-sided criterion for detection that allows for a maximum type error rate of $\alpha = 0.05$. Power by effect size was d = 0–0.464 (≤50%); d = 0.464–0.702 (50–80%); d = 0.702–0.930 (80–95); and >0.935 (≥95%). All statistical analyses were performed using SPSS version 20.0 for Windows (Chicago, IL, USA).

### 3. Results

The Shapiro–Wilk test of normality revealed that all data were normally distributed ($p > 0.05$). The t-test revealed no significant differences between the FP and LP in performance in fast maximal dynamic strength (A) (FP: 0.27 ± 0.04 m; LP: 0.29 ± 0.04 m; t = 0.98; $p = 0.34$; ES = 0.26), maximal dynamic strength (B) (FP: 88.14 ± 12.60 kg; LP: 87.07 ± 12.51 kg; t = 0.90; $p = 0.38$; ES = 0.24), and muscular endurance (C) (FP: 459.43 ± 92.71 s; LP: 456.93 ± 110.68 s; t = 0.27; $p = 0.78$; ES = 0.07) (Table 1).

**Table 1.** Comparison of performance between FP and LP.

| | | FP | | LP | | | | | |
|---|---|---|---|---|---|---|---|---|---|
| | | Mean ± SD | CI (95%) | Mean ± SD | CI (95%) | *t* | *p* | ES (d Cohen) | Interpretation |
| A | Jump height (m) | 0.27 ± 0.04 | 0.25–0.29 | 0.29 ± 0.04 | 0.24–0.29 | 0.98 | 0.34 | 0.26 | Trivial |
| | Velocity (m/s) | 1.16 ± 0.09 | 1.11–1.20 | 1.14 ± 0.09 | 1.08–1.20 | 1.08 | 0.30 | 0.28 | Trivial |
| | Force (N) | 1101.48 ± 133.62 | 1024.33–1178.63 | 1080.49 ± 124.58 | 1008.56–1152.42 | 1.71 | 0.11 | 0.45 | Small |
| | Power (W) | 1275.37 ± 200.42 | 1159.65–1391.09 | 1249.03 ± 207.40 | 1120.28–1359.78 | 1.26 | 0.22 | 0.33 | Trivial |
| B | 1-RM (kg) | 88.14 ± 12.60 | 80.86–95.42 | 87.07 ± 12.51 | 79.84–94.30 | 0.90 | 0.38 | 0.24 | Trivial |
| C | Karen® (s) | 459.43 ± 92.71 | 405.90–512.96 | 456.93 ± 110.68 | 393.02–520.83 | 0.27 | 0.78 | 0.07 | Trivial |

FP: follicular phase; LP: luteal phase; A: fast maximal dynamic strength; B: maximal dynamic strength; C: muscular endurance; SD: a deviation; CI: confidence interval; *t*: *t* student; *p*: significance index; ES: effect size.

## 4. Discussion

No significant differences were observed in performance in fast maximal dynamic strength, maximal dynamic strength, and muscular endurance between the FP and LP, corresponding with the initial hypothesis.

The participants in this investigation were well-trained athletes (at least 12 months of performing CrossFit®) with experience in all tests that were carried out during the assessments. This is important as there was no systematic learning effect between sessions, as was reported by other study, where participants enhanced their performance due to familiarization [37].

Our findings are consistent with several studies that reported no differences in strength and power performance throughout the different phases of the MC, even if oral contraceptives were used [23,38,39].

There is a lack of studies analyzing the effects of the MC on fast maximal dynamic strength using CMJ performance, and conflicting results have been reported with other jump assessments. For example, one study did not find differences in the squat jump (SJ) nor in the drop jump (DJ) when performance during the FP and ovulation phase were compared in active women [27]. One possible explanation is that women suffering from perimenstrual symptoms, such as having decreased energy and pain, might perform differently due plasma estradiol levels decreasing over time at a greater rate [40].

However, our study coincides with the available scientific literature, indicating that there are no changes in performance throughout the MC [41,42]. In Dasa et al. (2021), who studied high-level team sport players, no differences were seen between different MC phases during CMJ assessments [38]. For a better understanding of the CMJ performance, we studied jump height (m), mean velocity (m/s), mean force (N), and mean power (W), but we have not found significant differences in these variables.

Regarding maximal dynamic strength, we have not found differences in performance. Similar results were reported in one study that investigated the fluctuations of muscle performance regarding the Smith machine half-squat exercise during three different phases of the MC (early FP and FP, and mid-LP), reporting no differences in muscle strength and power performance [43]. This could be important for training prescription with the aim of adapting loads to the needs of athletes and optimizing adaptations. However, analyzing the effect of the MC in estimating 1-RM for the bench press exercise using the load–velocity relationship seems to not vary over the different phases of the MC [39].

On the other hand, regarding training neuromuscular adaptations, FP-based training showed a higher gain in muscle strength and diameter in type II fibers than LT-based

training. In addition, total testosterone and free testosterone were higher during the FP [7], with elite female athletes with higher salivary testosterone concentrations experiencing menstrual changes, leading to greater motivation and neuromuscular power [44]. This could be explained due to estradiol reaching its peak concentration in the late LP and enhanced cortical excitation, which has a positive association with excitability and muscle strength [45]. However, a recent study did not find a correlation between the individual variations in circulating estradiol, progesterone, or testosterone regarding performance in the CMJ, handgrip test, elbow flexor strength, and Wingate bike test. Furthermore, they did not observe differences in the physical or psychological well-being of the participants based on the phases of the MC [46].

No changes were found between the FP and LP regarding muscular endurance performance. Our results are consistent with the findings published in other studies, where the performance response to high-intensity interval exercise in athletes and oxygen consumption, energy expenditure, and relative perceived exertion were unaltered in the different phases of the MC [47]. These data are consistent with those reported by another study, where no differences were found in maximum heart rate, oxidation of fatty acids and carbohydrates, and even lactate concentrations [48].

Inconsistencies across studies may be observed due to different training methods being used, different types of muscular contractions being performed, and different muscle groups being engaged [25]. However, some female athletes perceive their performance to be relatively worse during early follicular and late luteal phases [24]. However, one study reported that for better performance during competition, elite female athletes identified that a preferred competition window usually occurs "just after your period" [49]. Despite the perceptual differences of athletes, no evidence has been found regarding the changes in the biomarkers of muscle damage based on the menstrual cycle [50]. Consequently, feelings and self-perceptions should be considered for some female athletes, providing control over predictability, timing, and symptoms of their MCs.

There are some limitations to this study that should be noted, such as the MC phases being detected with a urinary ovulation detection kit and not using hormonal blood tests. In addition, a higher sample size as well as monitoring the MC and performance over longer periods of time and cycles could have yielded different results.

Future research could focus on neuromuscular assessments using force plates and evaluate disciplines with a greater involvement of the cardiovascular system. In addition, studies could assess women without experience or with low physical conditioning to determine whether personal perceptions may have a greater influence in this population group.

## 5. Conclusions

CrossFit® performance in fast maximal dynamic strength, maximal dynamic strength, and muscular endurance is not altered during the MC in female CrossFit® athletes. However, monitoring the MC and its symptoms could be used to adapt athletes' training to their needs and preferences.

**Author Contributions:** Conceptualization, J.S.M.-S. and D.A.A.-A.; methodology, J.S.M.-S. and D.A.A.-A.; software, D.A.A.-A. and I.M.-G.; validation, D.A.A.-A., I.C.-M. and I.M.-G.; formal analysis, D.A.A.-A. and I.M.-G.; investigation, J.S.M.-S. and D.A.A.-A.; resources, D.A.A.-A.; data curation, J.S.M.-S. and D.A.A.-A.; writing—original draft preparation, D.A.A.-A. and I.M-G.; writing—review and editing, J.S.M.-S., D.A.A.-A., I.C.-M., and I.M.-G.; visualization, D.A.A.-A.; supervision, D.A.A.-A.; project administration, D.A.A.-A. All authors have read and agreed to the published version of the manuscript.

**Funding:** This research received no external funding.

**Institutional Review Board Statement:** This study was conducted in accordance with the Declaration of Helsinki and approved by the Ethics Committee of the Camilo José Cela University, Spain (approval code 16_23_RNM_FP and approval date 21 July 2023).

**Informed Consent Statement:** Informed consent was obtained from all subjects involved in this study.

**Data Availability Statement:** The data presented in this study are available on request from the corresponding author. The data are not publicly available due to privacy.

**Conflicts of Interest:** The authors declare no conflicts of interest.

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
