# Peer review of "Influence of the Menstrual Cycle on Performance in CrossFit®"

_applsci, doi:10.3390/app14135403_

Round 1
Reviewer 1 Report
Comments and Suggestions for Authors
In this paper, the authors shared their findings related to muscular power, strength, and endurance in female CrossFit athletes and whether those values were affected by the menstrual cycle. I found this paper to be interesting, and I agree that there is a noticeable gap in the literature related to this population and the potential affect on performance the menstrual cycle may have. Overall, the paper is written soundly, but there are several areas where a rewrite or grammatical corrections are necessary. I will note specific comments below and refer to the line number in the document draft.
- Line 21 -- I would like to see an explanation, perhaps not in Abstract but in the body, of why the 5-7 was FP and why LP was 20-23. There appears to be a greater range for LP than FP - why? Any why were these the two focal points related to the menstrual cycle? Would weekly testing have resulted in any differences?
- Line 57 -- Spelling of "oestrogen" here and throughout. Not sure if it needs to be corrected to "estrogen" prior to publication. I defer to editorial staff.
- Line 63 -- You mention that physical training can be improved but don't specify at which portion of the MC you are referring to. Provide a bit more clarity in this section and provide supporting evidence of why you would anticipate improvements.
- Line 64 -- Remove "'s" from female
- Line 70 -- What does the "..." denote? Either remove or replace with "etc."
- Line 109 -- When reading the "study design" section, I kept asking myself methodological questions, which ended up being answered on the next page. I would consider moving items from section 2.1.1 (see comment below) to section 2.2 so the procedures coincide with when the tests are introduced.
- Line 120 -- Were any participants regular consumers of caffeine prior to the study? If so, would the exclusion of this impact performance?
- Line 137 -- First, "2.1.1" seems like incorrect ordering since it follows section 2.2. Additionally, section 2.1 is already called "Participants". Did you mean to say "Procedures"? Either way, see comment above about combining 2.2 and 2.1.1 into a single section.
- Line 211 -- Table 1 appears to show all values goin down in LP with the exception of jump height. Since power went down, despite the increase in height, I was curious if body mass was recorded at each time point. You mention anthropometric data, but it appears to be baseline values. Since jump height went up but power went down, it would be useful to also share changes in body mass and explain why you think that was the case.
- Line 212 -- Please consider editing the document so the entire table appears on a single page. If this cannot be done, I recommend adding column headers to page 7 portion of the table.
- Line 213 -- were there any subjective assessments from FP to LP regarding fatigue, motivation, etc.?
- Line 218 -- What was the average amount of CF experience?
- Line 237 -- The sentences here need to be re-written. It was a bit confusing to read.
- Line 266 -- "be bote"?
Comments on the Quality of English Language
Overall, the paper is written with good grammar. There a couple areas where sentence structure or grammar issues were present, and I have addressed them above.
Author Response
Thank you very much for your contributions in the review carried out. We greatly appreciate your comments so that we can improve our manuscript. We have made appropriate adjustments to accommodate your comments including changes to the introduction, methodology, results, discussion and conclusions.
We attach a file with responses to your comments.
Kind regards.

Reviewer 2 Report
Comments and Suggestions for Authors
If you are aiming to assess neuromuscular aspects of sports, CMJ alone is not a valid marker. For neuromuscular assessment it would have been necessary for example to include force-plates that would have given you a better picture on inter-limb-asymmetries. If you use jump height and strength alone this reflects performance alone, not neuromuscular performance.
Introduction
Line 45: Why is it of interest? Please explain!
Line 62: Please specify how training can be improved? Improved regarding to what quality? Injuries? Performance?
Lines 68-69: This part is lacking logic. Are you referring to regular MC (which is assumed in terms of text before) or menstrual disturbances? Please specify and clarify. IF you want to refer to both, you need to explain it in more detail.
Line 77: What kind of adaption?
Lines 79-82: Do you consider neuromuscular performance and muscular contractility to be the same?
Methods
Please explain your sample size and include your statistical power under 2.3
Unfortunately, you fail to present neuromuscular data but have picked pure strength and endurance parameters.
Conclusion
Your title refers to neuromuscular performance, bur your conclusion and assessment fail to do so. Please make up your mind what athletic characteristics you want to analyze and revise your manuscript accordingly.
Discussion
Due to the lacking logic of the above aspects, it is nearly impossible to review this chapter.

Your manuscript clearly needs some thorough language edition and revision. Your sentence structure and sentence length make the text difficult to understand. The text does not flow smoothly and the reviewers are unable to fully assess your work in this form.
We have marked several aspects that need to be revised.
Author Response

(The authors gave the same response as above.)

Reviewer 3 Report
Comments and Suggestions for Authors
Authors present a study which evaluate physical performance of female athletes in 2 phases of menstrual cycle (FP and LP) and found no difference in the performance.
I found several major and minor problems.
1. Rationale for hypothesis is not properly described. It is not clear from the introduction why performance in FP and LP phases should be the same.
2. Rationale for the study needs to be better articulated. If there is systematic review (ref 28) of the studies similar to the presented by authors, the authors need provide a good explanation how exactly their study is different and what it aims to add.
3. In discussion, authors need to discuss merits of the finding in the light of multiple studies of similar nature.
4. Methodology / Results. Since both FP and LP phases are quite long intervals, authors need to introduce a measure to describe how measurements are distributed within the phases.
Minor
(see English)
Comments on the Quality of English Language1. line 61: "de" should be "a/the"
2. line 91: re-write the sentence "It is hypothesised ... " so that it is clear that it is hypothesis of this study / proposed by the authors
3. line 266: "bote" should be "noted?"
Author Response

(The authors gave the same response as above.)

Reviewer 4 Report
Comments and Suggestions for Authors
Differentiation of physical load in the process of constructing a training program is an important component of increasing its effectiveness. The subject of this study was to identify criteria for differentiating the training process for women in CrossFit.
This research addresses a little-studied area of knowledge about the characteristics of strength capabilities in women engaged in CrossFit during different phases of the menstrual cycle.
The authors mostly focus on scientific studies of recent years. There is no excessive self-citations, but at the same time we wanted to clarify the authors' experience in the topic under consideration.
The construction of the stages of the pedagogical experiment, research methodology meet the put forward hypothesis of the study and allow to reproduce the given pedagogical experiment.
The authors presented the results of the study in a sufficiently complete manner with further generalisation in the conclusions. The limitations of the article described by the authors could indeed expand the data obtained (blood tests for hormone levels, the number of participants in the pedagogical experiment, different levels of their fitness and other).
The relevance of the study is expanded by analysing the data of scientific literature on the peculiarities of the training process of women on the basis of taking into account the periods of the menstrual cycle. It is recommended to familiarise with the results of other authors on the topic: https://www.mdpi.com/2075-4663/12/1/31
The authors do not provide information in the introduction why Crossfit training was chosen as the object of the study.
Line 101 the author should clarify the wording " good technique".
Clarify whether oral contraception was an exclusion criterion.
Line 114-115 The author should clarify the duration of the study for participants whose menstrual cycle length was different from 28 days. Accordingly, clarify the onset of each phase.
Clarify whether the study participants were professional Crossfit athletes who actively participate in competitive activities or who exercise for recreational purposes.
When describing the Fast maximal dynamic strength, it is necessary to specify the conditions of video recording, whether the video camera was fixed, the frequency of video recording (frames per minute), whether it was a high-speed video recording, who recommended this test to evaluate the Fast maximal dynamic strength, the methodology of evaluation of this test (why the average value of three jumps was analysed).
For Figures 2, 4, give explanations for each of the images, what phase of the motor action is represented. Figure 2 (the moment of registration of a jump by the researcher) there is a question about observing the distance of 1.5 metres from the camera to the athlete (conclusion based on the average height of female athletes: 1.62±0.04 m), why a tripod was not used to fix the device for video recording.
To specify on what index the estimation of the Maximal dynamic strength was carried out, units of measurement.
In the conclusions, the authors should clarify the wording " highly trained women", how exactly was their level of fitness determined? Supplement the conclusions with numerical information and the results of statistical analysis.
Clarify the content of lines 288-290.
Author Response

(The authors gave the same response as above.)

Reviewer 5 Report
Comments and Suggestions for Authors
The authors aim to compare the performance of strength, power, and endurance between the follicular and luteal phases in well-trained CrossFit athletes. While this investigation could be relevant in this understudied area, the paper lacks clarity in several aspects, which obscures the key messages and potential applications. Specifically, the writing is often unclear, making the review challenging. Certain paragraphs in the introduction and discussion sections require rewriting for better comprehension. Additionally, I recommend that the authors seek assistance from a professional editing service specializing in helping authors write for publication in English.
Specific suggestions
The authors must clarify the terms neuromuscular performance, fast maximal dynamic strength, maximal dynamic strength, and muscular endurance throughout the manuscript. Understanding these concepts provides a foundation for interpreting the results of this study effectively.
I suggest the use of strength, power and muscle endurance or CrossFit performance.
The three paragraphs from lines 61 to 86 must be reordered and rewritten to maintain the relative flow within the text. For example, the information on relative strength (lines 72-75 and 79-81) should be introduced earlier in the text. By rearranging and revising these paragraphs accordingly, the coherence and logical progression of ideas can be enhanced. This adjustment ensures that the key concepts and supporting details are presented in a more organized and comprehensible manner, thereby improving the overall clarity and effectiveness of the text.
Line 88 delete hypothetically
Line 132 please define the order of the assessments.
By the end of the introduction, the last paragraph is not enough informative and can’t link the introduction with the aim of the study. This part of the introduction must be reorganized.
Line 227-230 please delete “other variables such as”
Line 231-248 Unstructured presentation of others' results. Please rewrite this pargraph, with a clearer point
Line 267 change “or” to “for” (instead of).
Line 269-272 Long and confusing sentence. please break down in shorter sentencesComments on the Quality of English Language
I suggest that the authors engage the assistance of a professional editing service that specializes in helping authors write for publication in English.
Author Response

(The authors gave the same response as above.)

Round 2
Reviewer 3 Report
Comments and Suggestions for Authors
I have no further suggestions or criticism.
Reviewer 4 Report
Comments and Suggestions for Authors
Good luck in further scientific research